

# Available potential energy of the three-dimensional mean state of the atmosphere and the thermodynamic potential for warm conveyor belts

Charles G. Gertler[1], Paul A. O'Gorman[1], and Stephan Pfahl[2]

[1]Department of Earth, Atmospheric, and Planetary Sciences, Massachusetts Institute of Technology, Cambridge, Massachusetts 02139, USA
[2]Institute of Meteorology, Freie Universität Berlin, 12165 Berlin, Germany

**Correspondence:** Charles G. Gertler (charles.gertler@gmail.com)

**Abstract.** Much of our understanding of the atmospheric circulation comes from relationships between aspects of the circulation and the mean state of the atmosphere. In particular, the concept of mean available potential energy (MAPE) has been used previously to relate the strength of the extratropical storm tracks to the zonal-mean temperature and humidity distributions. Here, we calculate for the first time the MAPE of the three-dimensional time-mean state of the atmosphere including the ef-
fects of latent heating. We further calculate a local MAPE by restricting the domain to an assumed eddy size, and we partition this local MAPE into convective and nonconvective components. Local nonconvective MAPE has a similar spatial pattern to the Eady growth rate, while local convective MAPE has some similarities in spatial pattern to a high percentile of instantaneous convective available potential energy. Furthermore, the maximum potential ascent associated with nonconvective local MAPE is strongly related to the frequency of warm conveyor belts (WCBs) which are ascending air streams in extratropical
cyclones with large impacts on weather. This maximum potential ascent can be calculated based only on mean temperature and humidity, and it also skillfully identifies the necessary conditions for WCBs at a given location on a specific day. These advances in the use of MAPE are expected to be helpful to connect changes in the mean state of the atmosphere, such as under global warming, to changes in important aspects of the extratropical circulation.

## 1 Introduction

The extratropical atmospheric circulation is dominated by baroclinic eddies that draw their energy from the mean available potential energy (MAPE) of the atmosphere. MAPE is the maximum amount of kinetic energy that could be released by reversible moist adiabatic motion starting from the mean state of the atmosphere, and it is calculated as the difference between the integrated enthalpy of the mean state and a hypothetical reference state (Lorenz, 1955, 1978). The reference state is the minimum-enthalpy state that can be attained through reversible moist adiabatic rearrangements. MAPE accounts for
the competing effects of the static stability, meridional temperature gradient, and moisture content on the energy available to circulations including extratropical cyclones.





Previous work has shown that the overall strength of the extratropical storm tracks as measured by eddy kinetic energy (EKE) is related to MAPE calculated from the zonal-mean state of the atmosphere. EKE scales linearly with MAPE in simulations with idealized climate models over a wide range of climates (Schneider and Walker, 2008; O'Gorman and Schneider, 2008; O'Gorman, 2011), in simulations with comprehensive climate models of global warming (O'Gorman, 2010) and solar geoengineering (Gertler et al., 2020), over the observed seasonal cycle (O'Gorman, 2010), and in the observed weakening trend of the Northern Hemisphere summertime storm track over recent decades (Gertler and O'Gorman, 2019). In targeted idealized simulations with isolated thermal forcings, the linear relation between EKE and MAPE can break down, but it still performs better in predicting changes in EKE than changes in meridional temperature gradient or static stability alone (Yuval and Kaspi, 2017). Thus MAPE calculated from the zonal-mean state has proven useful to connect aspects of storm-track behavior to the mean temperature and humidity.

MAPE is most often calculated over a hemisphere and allowing for any type of circulation, but restrictions have also been imposed on the parcel rearrangements in order to isolate particular contributions to MAPE in the atmosphere and ocean. For the atmosphere, a nonconvective MAPE has been proposed which does not permit vertical reordering of air originating at a given latitude, thereby limiting the release of convective instability (O'Gorman, 2010; Gertler and O'Gorman, 2019; Gertler et al., 2020). Nonconvective MAPE is thought to be more strongly related to large-scale EKE because kinetic energy generated by convective instability may be rapidly dissipated locally before any upscale energy transfer occurs (O'Gorman, 2010). A corollary to nonconvective MAPE, convective MAPE is defined as the difference between MAPE and nonconvective MAPE and is interpreted as the energy available for moist convection driven by large-scale circulations (Gertler and O'Gorman, 2019). For the ocean, an eddy-size constrained MAPE has been proposed in which no parcel may be displaced by a horizontal distance larger than the local eddy size (Su and Ingersoll, 2016). This eddy-size constraint takes into account that baroclinic eddies are generated by local parcel movement at the eddy length scale, and the spatial pattern of available potential energy density with the eddy-size constraint matches the spatial pattern of EKE in the Southern ocean (Su and Ingersoll, 2016).

In this paper, we calculate for the first time MAPE based on the three-dimensional state of the atmosphere including moisture, and we calculate the nonconvective and convective contributions to this three-dimensional MAPE. We also want to consider a local version of MAPE, but when we applied the eddy-size constraint approach of Su and Ingersoll (2016) to the atmosphere, the global optimization was overwhelmed by ascent in the deep tropics, where enthalpy is generally largest. As an alternative approach, we introduce a local MAPE that is calculated on a subdomain chosen to have a horizontal extent similar to that of a typical extratropical cyclone. By restricting parcel motions to a subdomain, local MAPE imposes a horizontal constraint on parcel motions that is similar to the eddy-size constraint of Su and Ingersoll (2016) but doesn't allow the ascent to be dominated by the tropics, and we investigate the properties of local MAPE including both nonconvective and convective local MAPE.

We furthermore explore how the parcel rearrangements involved in calculating local MAPE can provide insight into the dynamics of the extratropical storm tracks. We show here that these parcel rearragements can be used to identify a thermodynamic potential for warm conveyor belts (WCBs). WCBs are strongly ascending air streams that originate in the atmospheric boundary layer in the warm sector of an extratropical cyclone and typically move polewards while ascending to levels near the tropopause. WCBs play a large role in cloud and precipitation formation and in transport of energy and air pollution (Brown-



ing, 1990; Stohl et al., 2002), and they are responsible for many extreme precipitation events in the extratropics (Pfahl et al., 2014). More than half of extratropical cyclones are associated with a WCB in northern hemisphere winter, while the fraction of cyclones associated with a WCB is much lower in northern hemisphere summer when the atmosphere is moist but EKE
is weak, and cyclones near the coast of Antarctica are rarely associated with a WCB presumably because of low humidity in that region (Madonna et al., 2014). Here, we make use of the WCB climatology of Madonna et al. (2014) which identifies WCBs as Lagrangian trajectories of air parcels that occur in the vicinity of an extratropical cyclone and experience ascent of more than 600 hPa within 2 days. This is a relatively strict criterion that identifies strong ascent involving moist processes. We demonstrate that regions of frequent WCBs correspond to regions of large ascent between the mean state and the reference
state in the calculation of nonconvective local MAPE, thus making a link between WCB formation and the mean temperature and moisture of the atmosphere.

One challenge in using MAPE is that the reference state can be difficult to calculate accurately when atmospheric moisture is taken into account. For the case of a dry atmosphere, Lorenz (1955) derived an approximate formulation of the available potential energy using an assumption of small isentropic slope that is reasonably accurate in the free troposphere but prob-
lematic in the boundary layer. In this approximate dry formulation, the dominant contribution to MAPE is from the zonal and time mean rather than stationary eddies (Oort et al., 1989; Peixoto and Oort, 1992), and this has been used to justify the two-dimensional calculation of MAPE based on the zonal mean state. For the case of a moist atmosphere, latent heating needs to be accounted for in the calculation of MAPE, and the determination of the reference state is more difficult because a conserved moisture variable needs to be accounted for in addition to entropy (Lorenz, 1978). Lorenz (1979) introduced a parcel
swapping algorithm to calculate the reference state of a moist zonal-mean atmosphere divided into equal mass parcels. Randall and Wang (1992) showed that the Lorenz (1979) algorithm does not always find the exact reference state, and they presented a different algorithm for solving for the reference state that succeeds in certain cases where Lorenz (1979) fails, but it similarly does not always solve for the exact reference state (Stansifer et al., 2017). Important advances were made recently when the first exact calculations of MAPE were made using the Munkres algorithm for the ocean (Hieronymus and Nycander, 2015) and
atmosphere (Stansifer et al., 2017). A MAPE calculation is said to be exact in this context when the exact MAPE is found for a given mean state and set of thermodynamic assumptions. Stansifer et al. (2017) also introduced an inexact divide-and-conquer algorithm which is fast and gives an accurate answer in most situations (Stansifer et al., 2017; Harris and Tailleux, 2018). In this paper, we frame the MAPE calculation following the formulation used for the ocean by Su and Ingersoll (2016) to take advantage of having many parcels at each pressure level, and we solve for the reference state using a standard integer linear
programming algorithm which is faster than Munkres for this problem and also exact. However, we also make use of the inexact divide-and-conquer algorithm for nonconvective MAPE because the integer linear programming algorithm does not always converge when the nonconvective restriction is included.

The paper proceeds as follows. In Section 2, we describe the reanalysis data and WCB climatology that are used in the paper. In Section 3, we describe the methods used to calculate the three-dimensional MAPE of the atmosphere, including
the imposition of nonconvective and local restrictions. In Section 4, we present results for three-dimensional atmospheric MAPE with and without the nonconvective and local restrictions. In Section 5, we relate the parcel ascent in the calculation





of nonconvective local MAPE to a WCB climatology and to daily WCB activity. In Section 6, we discuss the results, present brief conclusions, and propose avenues for future work.

## 2 Reanalysis data and WCB climatology

All MAPE calculations and comparisons to meteorological fields are based on the ERA-Interim reanalysis dataset, a global atmospheric reanalysis produced by the European Center for Medium-Range Weather Forecasts (Dee et al., 2011). We use temperature, relative humidity, wind, and surface pressure data. In most cases, we use monthly data at a horizontal grid spacing of 0.75° by 0.75° and at 37 pressure levels. However, for 6-hourly convective available potential energy (CAPE) and for daily MAPE, we use six-hourly data at a horizontal grid spacing of 2.5° by 2.5° and at 37 pressure levels. Unless otherwise noted,

all time means are over 1979-2018. Data at levels below the surface are removed.

In addition, we compare to the WCB climatology developed by Madonna et al. (2014) which has been extended in time since publication to cover the time period 1979-2018. WCB trajectories are identified in ERA-interim as strongly ascending air parcels (600 hPa in 2 days) that occur in the vicinity of an extratropical cyclone. A trajectory is deemed to be within the vicinity of an extratropical cyclone if it is within the horizontal area bounded by the outermost closed contour of sea level

pressure for at least one 6-hourly time step during the 2-day period of strong ascent.

## 3 Calculation of MAPE

This section describes the methods we use to calculate MAPE of the three-dimensional time-mean state of the atmosphere. The main challenge is to calculate the parcel rearrangement leading to the reference state (i.e., the minimum enthalpy state), and the MAPE is then calculated for a given domain as the integrated enthalpy of the mean state minus that of the reference

state. We also consider two constraints on the parcel rearrangement. The first constraint, used in calculating nonconvective MAPE, restricts the release of convective instability. The second constraint, used in calculating a local MAPE, restricts parcel movements to a local subdomain centered on a given surface location with a nominal radius in the horizontal chosen to represent a characteristic eddy length scale.

### 3.1 Dividing the atmosphere into equal-mass parcels

The global atmosphere is divided into parcels of equal mass to simplify the parcel rearrangement. In previous studies of MAPE based on the zonal mean (which we will refer to as zonal-mean MAPE), the domain in latitude and pressure was divided into equal mass parcels by using uniform spacing in pressure and a uniform spacing in a cosine-weighted latitude coordinate which gives equal area divisions in the horizontal (Lorenz, 1979; O'Gorman, 2010; Gertler and O'Gorman, 2019; Gertler et al., 2020). This approach has the disadvantage that the spacing in latitude increases towards the poles, and for the three-

dimensional calculation of MAPE it leads to strongly varying aspect ratios of grid cells that are particularly problematic for the calculation of local MAPE at high latitudes. Instead, we generate a uniform grid on the sphere (i.e., with equal-area grid



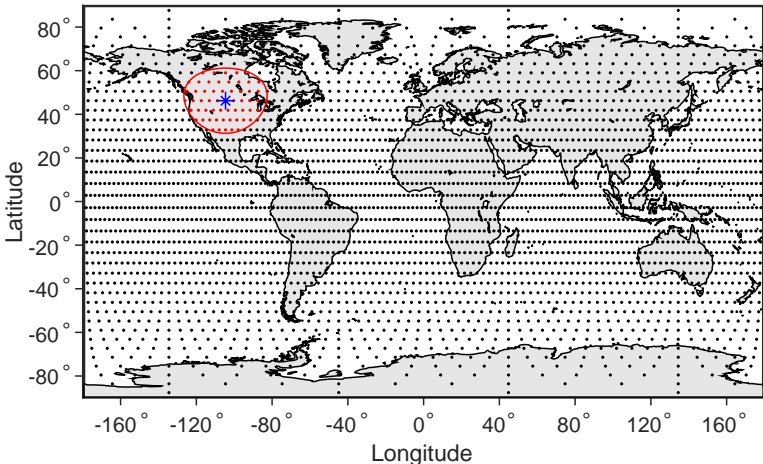

**Figure 1.** Example of a uniform global grid created using the methodology introduced in Rosca (2010). There are 3200 grid points based on a $40 \times 40$ grid in each hemisphere. The red circle is an example subdomain for the local MAPE calculation with a geodesic radius of 15 degrees (1670 kilometers) centered on the location marked with a blue asterisk and including all grid points colored in red. Except where noted, MAPE results in figures and tables are based on this horizontal grid at 15 vertical pressure levels.

cells in the horizontal) following the approach of Rosca (2010) as described in detail in Appendix A. An example of such a horizontal grid is shown in Figure 1.

Some previous studies of zonal-mean MAPE have used a staggered grid which reduces the problem to one dimension in pressure (Lorenz, 1979; O'Gorman, 2010; Gertler and O'Gorman, 2019), but it is equally simple and more efficient to employ the same pressure coordinates in all columns (Su and Ingersoll, 2016), and that is the approach used here, with uniform spacing in pressure to give equal mass parcels.

All results (except for sensitivity tests) are based on 15 evenly-spaced pressure levels in the vertical, and any parcels below the surface are excluded. We generally use a global grid with 3200 surface grid cells (a $40 \times 40$ grid in each hemisphere), except in the case of non-convective and convective MAPE of extratropical domains (20°-90°) which are calculated using a global grid with 800 surface grid cells (a $20 \times 20$ grid in each hemisphere) due to computational expense.

### 3.2 Determining the reference (minimum-enthalpy) state

We consider $n$ parcels of equal mass $M$, each with an initial temperature ($T_i$), pressure ($p_i$), and relative humidity ($r_i$) where $i = 1, 2, ..., n$. The parcels can occur at $s$ distinct pressure levels, $p_k^L$, where $k = 1, 2, ..., s$ and the superscript $L$ refers to level. The number of parcels at each level is denoted by $m_k$, and $m_k$ varies by level due to surface topography. Note that, by construction, $\sum_{k=1}^{s} m_k = n$. A matrix is defined which contains the enthalpy values of all parcels adiabatically and reversibly displaced to every pressure level, $\boldsymbol{h} = [h_{i,k}](i = 1, 2, ..., n; k = 1, 2, ..., s)$, where $h_{i,k}(T_i, p_i, r_i, p_k^L)$ is the specific enthalpy of parcel $i$ when it has been displaced to pressure $p_k^L$. In calculating the enthalpy of parcels at different pressures, we use the





saturation vapor pressure formulae over ice and liquid described in Simmons et al. (1999), but follow Wang and Randall
(1994) to merge ice and liquid phases. We then define the binary matrix $\boldsymbol{x} = [x_{i,k}](i = 1, 2, ..., n; k = 1, 2, ..., s)$ that maps the
mean state onto the reference state, where $x_{i,k} = 0$ or 1, and $x_{i,k} = 1$ indicates that parcel $i$ is located at pressure $p_k^L$ in the
reference state. A given rearrangement has a unique $\boldsymbol{x}$ matrix, and the total enthalpy of the reference state is $M \sum_{i=1}^{n} \sum_{k=1}^{s} h_{i,k} x_{i,k}$.
The sum over each row of $\boldsymbol{x}$ must equal one to ensure that each parcel is only assigned one location in the rearrangement state,
and the sum over each column of $\boldsymbol{x}$ must be the same as the number of parcels at the corresponding pressure level ($m_k$).
Therefore, to solve for the reference state, one finds $\boldsymbol{x}$ which minimizes the total enthalpy given these conditions. Following
Su and Ingersoll (2016), the problem can be stated as follows:

Given an $n \times s$ matrix $\boldsymbol{h}$, find an $n \times s$ matrix $\boldsymbol{x}$ to minimize $\sum_{i=1}^{n} \sum_{k=1}^{s} h_{i,k} x_{i,k}$,

where $x_{i,k} = 0$ or 1, subject to $\sum_{k=1}^{s} x_{i,k} = 1$ for any $i$ and $\sum_{i=1}^{n} x_{i,k} = m_k$ for any $k$. $\qquad$ (1)

Su and Ingersoll (2016) recognize that this problem is an example of a minimum-cost flow problem. It is also an integer
linear programming problem, and we solve it here using a standard implementation of the dual-simplex algorithm (Koberstein,
2008). To frame the problem for integer linear programming, the $\boldsymbol{h}$ and $\boldsymbol{x}$ matrices are vectorized to $\boldsymbol{h_{vec}}$ and $\boldsymbol{x_{vec}}$, and we
minimize $\boldsymbol{h_{vec}} \cdot \boldsymbol{x_{vec}}$ over $\boldsymbol{x_{vec}}$, subject to the constraint that $\boldsymbol{A} \cdot \boldsymbol{x_{vec}} = \boldsymbol{b}$, $\boldsymbol{A}$ and $\boldsymbol{b}$ being a matrix and vector, respectively,
that express the constraints in (1).

We also solve for the reference state using the divide-and-conquer algorithm described in Stansifer et al. (2017). While
the divide-and-conquer algorithm is slower than the integer linear programming approach, the algorithm used to solve the
integer linear programming approach occasionally does not converge under the nonconvective condition imposed on the parcel
rearrangement described below. The divide-and-conquer algorithm is a recursive algorithm that builds a low-enthalpy reference
state by dividing the atmospheric domain into smaller subdomains. At each division, the pressure-derivative of enthalpy at the
middle pressure of the subdomain is used to order parcels within that subdomain from top to bottom, and then the top and
bottom halves are assigned to new subdomains.

Previous work has shown that the reference pressure for the minimum enthalpy state of the zonal-mean atmosphere for the
moist MAPE calculation has a convective "bubble" at low latitudes that ascends discontinuously to the upper troposphere,
while for nonconvective MAPE the reference pressure is always continuous as a function of latitude and pressure, by definition
(see, e.g., Figure 2 of Gertler and O'Gorman (2019)). A similar area of low-latitude ascent in the three-dimensional moist
MAPE calculation can be seen in Figure 2, but now resolved in latitude and longitude.

### 3.3 Calculation of nonconvective and convective MAPE

For nonconvective MAPE, the reference state is calculated under the condition that parcels originating in a given column may
not change their vertical ordering, thereby restricting the release of convective instability. We calculate nonconvective MAPE
using the divide-and-conquer algorithm, and this is done by modifying the algorithm such that at each step of the recursion,





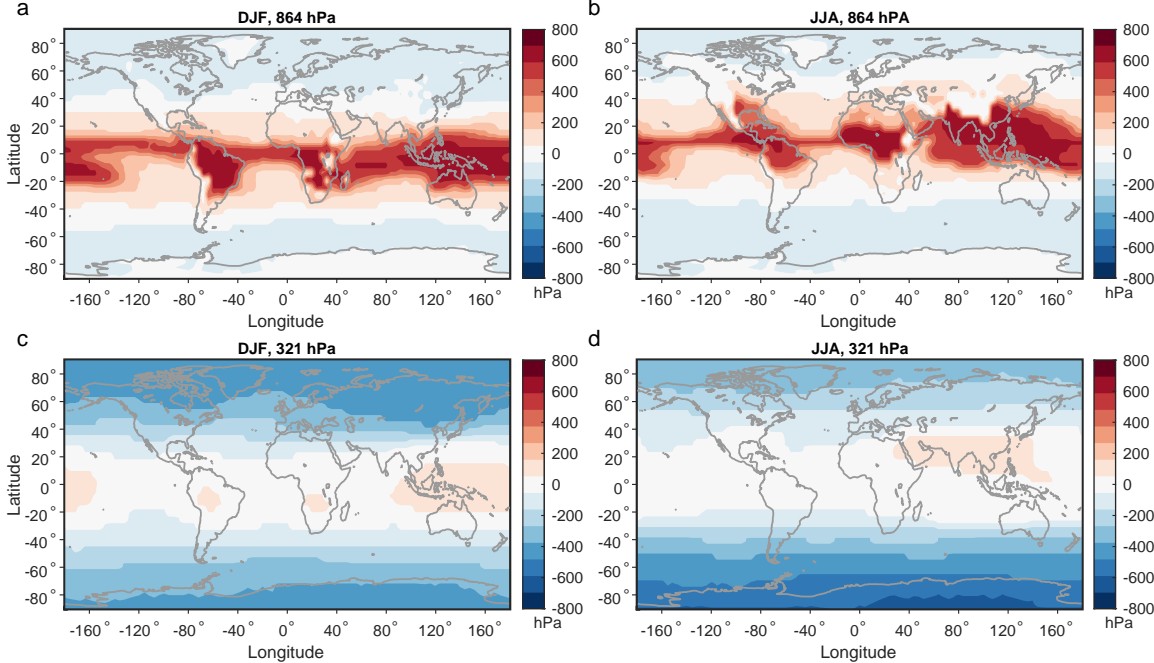

**Figure 2.** Visualization of vertical parcel motions in global three-dimensional MAPE calculations for different seasons. Shown is the original pressure minus reference pressure for parcels originating at (a,b) 796 hPa and (c, d) 321 hPa for (a, c) DJF and (b, d) JJA. Positive values imply ascent from the mean state to the reference state, and negative values imply descent from the mean state to the reference state.

parcels from a given initial column may not change their vertical ordering in the parcel rearrangement of each subdomain. This ensures that parcels do not "leapfrog" in pressure over parcels from the same initial column, and the reference pressure is a monotonic function of pressure in a given column. The nonconvective MAPE is then defined as the integrated enthalpy of the mean state minus that of the reference state restricted in this way.

The convective MAPE is defined as the difference between the MAPE and the nonconvective MAPE. When calculating convective MAPE, both the MAPE and the nonconvective MAPE are calculated using divide and conquer in the case of the extratropical domains (20°-90° in latitude), but for local MAPE we use divide and conquer for nonconvective MAPE and integer linear programming for MAPE because divide and conquer sometimes gave negative values in the tropics (see section 4.3.2 for further discussion).

## 3.4 Calculation of local MAPE

We perform a series of MAPE calculations within local subdomains whose horizontal extent is set to a nominal length scale of an extratropical cyclone. This approach captures the energy available locally to be converted into kinetic energy in an extratropical cyclone, but note that it does not account for horizontal energy fluxes into and out of these systems such as the





fluxes that occur in downstream development (Chang, 1992; Chang et al., 2002), and thus we expect regions of high local MAPE to correspond to regions of large generation of EKE, but we do not expect local MAPE to capture the downstream structure of the storm track.

As before, the atmosphere is divided into parcels of equal mass using a uniform grid on the sphere and equal divisions in pressure. At each surface location, a subdomain is defined as including only parcels that fall within a horizontal distance less than a specified eddy length scale of that location. The MAPE at a given surface location is defined as the MAPE over the subdomain centered in the horizontal on that location. An example of such a subdomain is shown in Figure 1. We consider the MAPE and its nonconvective and convective components in each subdomain, and record the reference pressures of parcels in 190 each MAPE calculation.

Using local MAPE as an example, we illustrate the sensitivity of the MAPE calculation to key parameters and the algorithm used in Appendix B.

## 4 Three-dimensional MAPE of the atmosphere

### 4.1 Extratropical and global MAPE

The seasonal MAPE over 1979-2018 is shown in Table 1 for northern and southern extratropical domains (20° -90° latitude) and for the global domain. We focus on the extratropical domains but give results for the global domain for reference. The results in Table 1 are based on a global uniform grid with 3200 equal-area surface grid cells (a $40 \times 40$ grid in each hemisphere), and 15 evenly spaced pressure levels from 1000 hPa to 50 hPa, with parcels below the surface removed. The results are generally similar beyond a resolution of about 800 surface grid cells and 10 pressure levels. The grid cells in this global grid that fall 200 within the extratropical domains are used to calculate the extratropical MAPE values. A zonal-mean MAPE on the same domain is also calculated by replacing all the temperature and humidity values with their zonal mean values at that latitude and pressure. We use this approach to calculating zonal-mean MAPE in which the only inputs are zonal-mean temperature and humidity but the calculation remains three-dimensional because it allows for a fair comparison with fully three-dimensional MAPE in the presence of topography that varies in longitude.

MAPE is larger in winter than summer (Table 1), implying that the effect of the stronger meridional temperature gradient in winter outweighs the lower specific humidity. The three-dimensional MAPE is always larger than the zonal-mean MAPE, and this is as expected because three-dimensional MAPE includes both zonal and meridional gradients of temperature and moisture. The difference between the three-dimensional MAPE and the zonal-mean MAPE, which can be interpreted as the contribution from zonal asymmetries, is always largest in the Northern Hemisphere extratropics, with particularly large contributions in DJF 210 and JJA. The contribution from zonal asymmetries in the Northern Hemisphere extratropics in JJA of 22.5% is substantially larger than what was found previously for dry MAPE for the full NH (7.5% based on Table 14.1 of Peixoto and Oort (1992)).

Figure 2 illustrates the vertical motion of parcels in the global three-dimensional MAPE calculations for DJF and JJA, showing tropical and subtropical ascent and extratropical descent. Tropical and subtropical ascent is strongest at low levels in the summer hemisphere and over land, and extratropical descent is strongest at the upper levels in the winter hemisphere.





**Table 1.** Extratropical and global MAPE values in J kg$^{-1}$ based on the time mean over 1979-2018 for different seasons. Results are given for three-dimensional MAPE, zonal-mean MAPE, and their difference. Zonal-mean MAPE is calculated by replacing all values for temperature and humidity at a given latitude and pressure with the zonal-mean values at that latitude and pressure. The difference is calculated three-dimensional MAPE minus zonal-mean MAPE as a percentage of the three-dimensional MAPE. All results in this table are calculated with integer linear programming.

| Season | Domain | MAPE (J kg$^{-1}$) | Zonal-Mean MAPE (J kg$^{-1}$) | Difference (%) |
|--------|--------|--------------------|-------------------------------|----------------|
| DJF | 20°-90° N | 373.6 | 335.3 | 10.2 |
| | 20°-90° S | 301.2 | 284.1 | 5.6 |
| | 90° S -90° N | 480.4 | 452.8 | 5.7 |
| MAM | 20°-90° N | 303.0 | 285.5 | 5.8 |
| | 20°-90° S | 333.0 | 324.7 | 2.5 |
| | 90° S -90° N | 450.6 | 435.9 | 3.3 |
| JJA | 20°-90° N | 215.2 | 166.8 | 22.5 |
| | 20°-90° S | 361.0 | 351.1 | 2.8 |
| | 90° S -90° N | 433.8 | 407.6 | 6.1 |
| SON | 20°-90° N | 324.8 | 305.7 | 5.9 |
| | 20°-90° S | 341.8 | 330.1 | 3.4 |
| | 90° S -90° N | 443.2 | 423.3 | 4.5 |

Nonconvective MAPE and convective MAPE, calculated using divide and conquer, are presented in Table 2, and these are calculated on a lower-resolution grid due to the greater computational requirement when the non-convective constraint is imposed and when using the divide-and-conquer algorithm. The global domain is divided into equal mass parcels using 1800 equal-area grid cells (a $30 \times 30$ grid in each hemisphere) and 15 evenly spaced pressure levels from the surface to 50 hPa. Using this grid and the divide-and-conquer algorithm for full three-dimensional MAPE gives very similar values to those from the
higher-resolution exact calculation using integer linear programming (compare the extratropical MAPE values in Tables 1 and 2).

Nonconvective MAPE restricts vertical reordering of parcels starting in a column, and thus it is is always equal or smaller in magnitude to the MAPE. However, MAPE and nonconvective MAPE are similar in magnitude with the largest values in Northern Hemisphere DJF and smallest values in Northern Hemisphere JJA. Convective MAPE is largest in Northern
Hemisphere JJA, when it is also the largest percentage of overall MAPE, and smallest in Northern Hemisphere DJF and Southern Hemisphere JJA and SON. The relative breakdown between convective and nonconvective MAPE across seasons and hemispheres are similar for zonal-mean MAPE as for three-dimensional MAPE (not shown).



**Table 2.** Extratropical three-dimensional MAPE, nonconvective MAPE, and convective MAPE in J kg$^{-1}$ over 1979-2018. The convective fraction is shown in parentheses as a percentage of full MAPE. Results in this table are based on a lower-resolution horizontal grid with 1800 surface grid cells (a $30 \times 30$ grid in each hemisphere), and the (full) MAPE values differ slightly from those in Table 1 because of the lower resolution and because divide and conquer is used here rather than linear integer programming in Table 1.

| Season | Domain | MAPE (J kg$^{-1}$) | Nonconvective MAPE (J kg$^{-1}$) | Convective MAPE (J kg$^{-1}$) |
|---|---|---|---|---|
| DJF | 20°-90° N | 372.5 | 366.9 | 5.7 (1.5%) |
| | 20°-90° S | 300.7 | 285.2 | 15.4 (5.1%) |
| MAM | 20°-90° N | 303.8 | 296.1 | 7.7 (2.5%) |
| | 20°-90° S | 333.2 | 318.8 | 14.4 (4.3%) |
| JJA | 20°-90° N | 216.7 | 197.3 | 19.4 (9.0%) |
| | 20°-90° S | 359.1 | 353.4 | 5.7 (1.6%) |
| SON | 20°-90° N | 326.1 | 309.8 | 16.4 (5.0%) |
| | 20°-90° S | 341.4 | 335.8 | 5.6 (1.7%) |

## 4.2 Local MAPE

Next the local restriction is imposed on the three-dimensional MAPE calculation. We use the same global grid as for the full three-dimensional MAPE calculation: 3200 equal-area surface grid cells (a $40 \times 40$ grid in each hemisphere) and 15 evenly spaced pressure levels from the surface to 50 hPa. Each subdomain for the local MAPE is centered in the horizontal at one surface grid cell and has a geodesic radius of 15 degrees (1670 kilometers). This radius was chosen as representative of the length scale of midlatitude eddies.

The local MAPE based on the climatological mean over 1979-2018 is shown in Figure 3. Local MAPE represents the energy available for local parcel rearrangement at the eddy length scale, and large values correspond to extratropical regions with strong baroclinic instability. The eastern sides of continents and the western side of ocean basins are particular hotspots. The Northern Hemisphere has the highest values in the Western Pacific in winter, while the Southern Hemisphere shows more consistently high values throughout the midlatitudes in winter. By restricting to subdomains that are roughly of the size of a cyclone, we have effectively introduced more dynamics into the calculation of MAPE and reduced its value to be more realizable and similar in magnitude to kinetic energy. For example, in DJF, the local MAPE averaged over the northern extratropics is 96.8 J kg$^{-1}$ as compared to 373.6 J kg$^{-1}$ for MAPE calculated for the northern extratropics without the local constraint.

The reference pressure distribution for each subdomain may be examined for insight into the characteristics of the local parcel rearrangements. In particular, the difference between the original pressure and reference pressure of a parcel may be interpreted as the ascent or descent that gives rise to the potentially largest energy release on the eddy length scale. Parcels in the column above a given surface location are included in multiple MAPE calculations for subdomains that overlap with





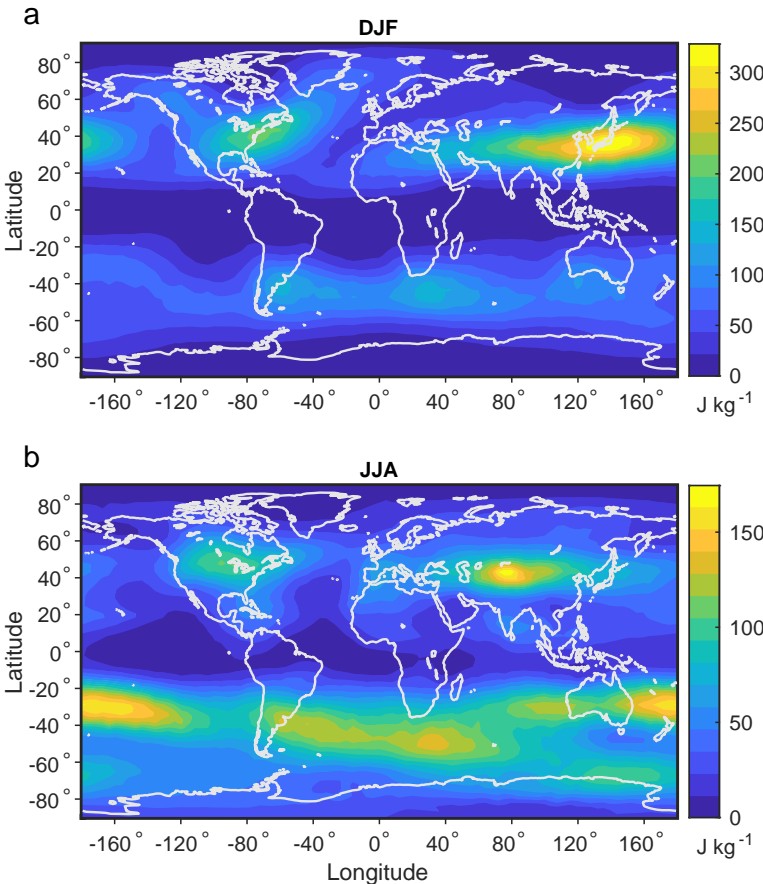

**Figure 3.** Local MAPE based on the climatological mean over 1979-2018 for (a) DJF and (b) JJA. MAPE is calculated on subdomains of geodesic radius 15 degrees (1670 kilometers) centered in the horizontal on 3200 surface grid cells.

that surface location. Therefore, we calculate the maximum ascent of any parcel over all subdomains that include that surface location, and this can be thought of as the maximum potential ascent due to baroclinic instability and convection at that location. Figure 4 shows the maximum potential ascent in each season, with maximum values near the equator and high values over midlatitude land in summer. Much of this is convective ascent, and the nonconvective and convective components of local MAPE are next examined separately.




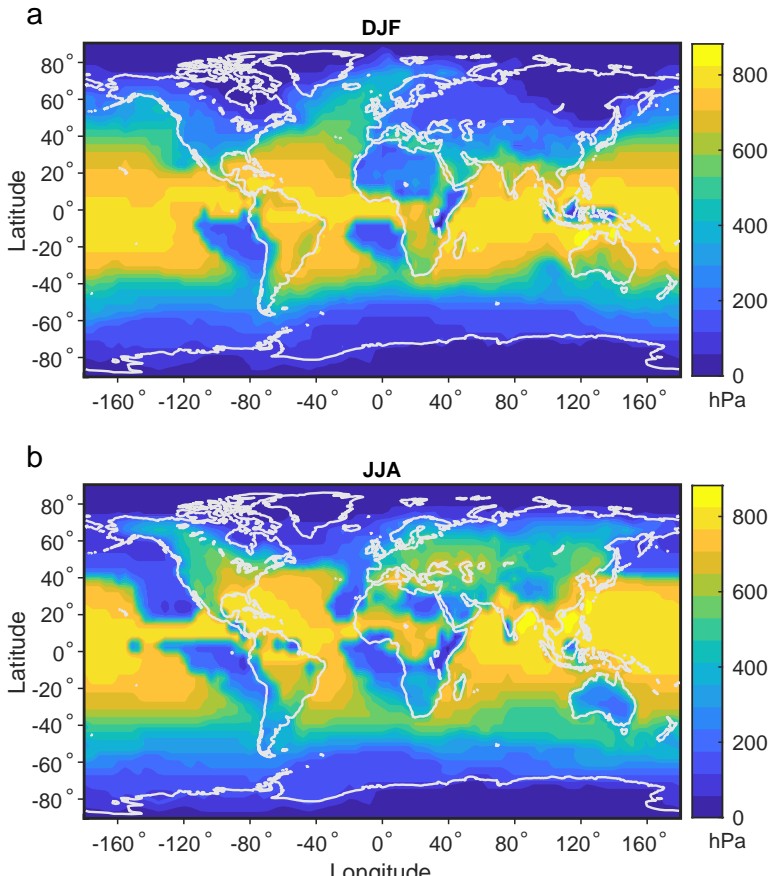

**Figure 4.** Maximum potential ascent in the local MAPE calculation based on the climatological mean over 1979-2018 for (a) DJF and (b) JJA. MAPE is calculated as in Figure 3, and the maximum potential ascent is calculated at each surface location as the maximum difference between pressure and reference pressure for any parcel in the column at that location over all the MAPE calculations that include that location.

## 4.3 Nonconvective and convective local MAPE

### 4.3.1 Nonconvective local MAPE

The nonconvective local MAPE is also calculated using the same grid and subdomains as for the full local MAPE, but using the divide-and-conquer algorithm rather than integer linear programming. Nonconvective local MAPE must be smaller or equal to local MAPE, but Figure 5 shows that the overall pattern and magnitude of nonconvective local MAPE are similar to those of local MAPE in Figure 3.





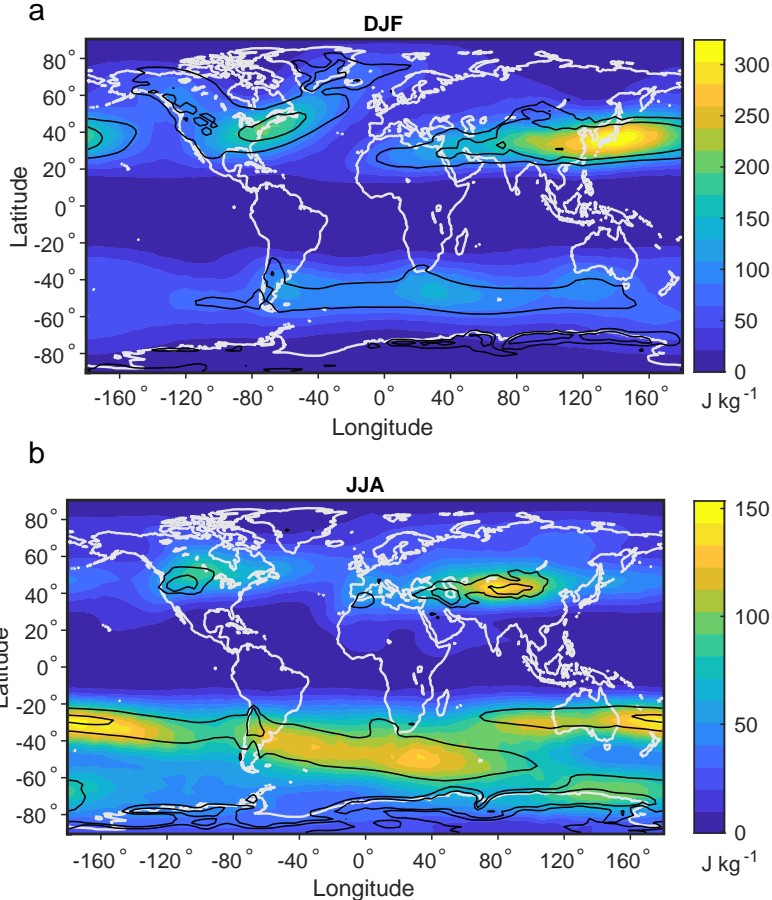

**Figure 5.** Nonconvective local MAPE (color shading) and Eady growth rate averaged in the vertical over 500-750 hPa (black contours of 0.5 and 0.7 day$^{-1}$) based on the climatological mean over 1979-2018 in (a) DJF and (b) JJA.

Nonconvective local MAPE has high values in the entrance regions of the storm tracks, and we interpret it as a measure of the available energy for the generation of large-scale eddies through moist baroclinic instability. Dry MAPE scales roughly
as the vertical integral of the square of the Eady growth rate (O'Gorman and Schneider, 2008), and Figure 5 shows that the nonconvective local MAPE has a similar spatial pattern to the Eady growth rate averaged in the vertical over 500-750hPa. The (maximum) Eady growth rate, $\sigma$, at a given vertical level is calculated as

$$\sigma = 0.31 \frac{f}{N} \left\| \frac{\partial \mathbf{u_h}}{\partial z} \right\|, \tag{2}$$

where $f$ is the Coriolis parameter, $N$ is the buoyancy frequency, $\mathbf{u_h}$ is the horizontal wind, and $z$ is height. Local MAPE is
complementary to the Eady growth rate as a measure of local baroclinic instability in that (1) it provides a measure of energy that can be compared to EKE in terms of fractional changes over the seasonal cycle or for climate change, and (2) it includes





the effects of moisture which can be important in terms of the response to climate change (although we note the Eady growth rate could also be modified to account for moisture through an effective static stability (O'Gorman, 2011)).

### 4.3.2 Convective local MAPE

Figure 6 shows convective local MAPE which is defined as the difference between full local MAPE and nonconvective local MAPE. The full local MAPE is calculated using the integer linear programming approach because the divide-and-conquer algorithm gives some negative values near the equator. Negative values of MAPE are physically impossible, but they can sometimes occur with the divide-and-conquer algorithm for conditionally unstable atmospheres because the divide-and-conquer algorithm is not exact (Harris and Tailleux, 2018). Thus, integer linear programming is used to calculate the full local MAPE while divide

and conquer is used to calculate the nonconvective local MAPE, and we caution that this mixing of algorithms could affect the results for convective local MAPE in localized regions near the equator (see Appendix B, section B3).

The convective local MAPE is interpreted as a local measure of the energy available for convection associated with baroclinic eddies in the atmosphere. To compare it to a conventional metric of convective potential, we calculate CAPE using vertical profiles of six-hourly temperature and humidity fields and using the same moist thermodynamic formulation as in

the MAPE calculation for consistency. CAPE is defined as the positive kinetic energy generated by a surface air parcel lifted through the atmosphere from the level of free convection to the level of neutral buoyancy (Emanuel, 1994). The $95^{th}$ percentile of six-hourly CAPE is plotted in Figure 6 over the same time period as the climatological convective MAPE, comparing a quantity derived from instantaneous values to a quantity derived from the time-mean state of the atmosphere. Note that MAPE values are generally much smaller than CAPE values because MAPE is normalized by the full mass of the atmosphere whereas

CAPE is normalized by only the mass of the air that is rising. Figure 6 demonstrates that the poleward extent of high convective MAPE is similar to the poleward extent of high values of the $95^{th}$ percentile of CAPE. However, convective MAPE exhibits a local minimum near the equator because it represents the convection associated with large-scale circulations driven by horizontal temperature gradients which are small near the equator. This local minimum is absent for the $95^{th}$ percentile of CAPE, presumably because CAPE can additionally be driven by short-term (sub-daily to daily) variations in other factors such

as surface fluxes not accounted for in the MAPE framework. The maximum potential ascent in the calculation of local MAPE does have large values near the equator, and this is the case even if the maximum potential ascent is reported at the center of the subdomain in which it occurs rather than at the location of the ascent as in Figure 4. However, large displacements in the vertical do not imply large generation of kinetic energy if the buoyancy is low, explaining how convective MAPE values can remain small in that region. Further study is needed to better connect convective MAPE to instantaneous atmospheric

convection, perhaps using an idealized modeling approach.




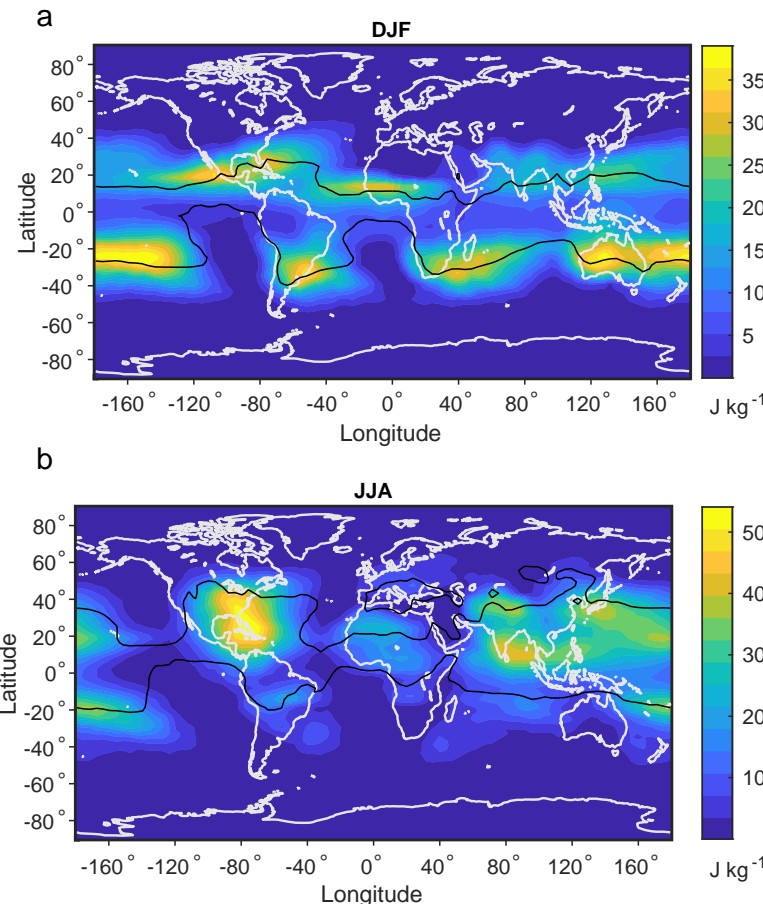

**Figure 6.** Convective local MAPE based on the climatological mean (color shading) and $95^{th}$ percentile of six-hourly CAPE (black contour of 1500 J kg$^{-1}$) over 1979-2018 for (a) DJF and (b) JJA. Convective MAPE is calculated as the difference in MAPE and nonconvective MAPE in the local subdomains.

## 5 Application to Warm Conveyor Belts (WCBs)

### 5.1 Climatological maximum potential ascent and WCBs

Maximum potential ascent determined by the local MAPE framework can provide insight into WCB activity given that WCBs are instances of strong atmospheric ascent associated with extratropical cyclones. Ascent in WCBs is strongly diabatic and thus it is important that we are using a moist MAPE that accounts for latent heating. High-resolution simulations and satellite observations suggest that WCBs trajectories are mostly slantwise but do include instances of more vertical convective ascent embedded in the slower slantwise ascent (Rasp et al., 2016; Oertel et al., 2019, 2020). However, we focus here on ascent in






calculations of nonconvective MAPE (excluding convective ascent) rather than full MAPE because we find it gives a stronger connection with WCBs, and this is presumably because the full MAPE calculation also includes convective ascent not associ-

ated with extratropical cyclones (e.g., at low latitudes).

Figure 7 shows the maximum potential ascent for the local nonconvective MAPE described in Section 4.3.1. The strong ascent in equatorial regions for the full local MAPE in Figure 4 is absent for the nonconvective local MAPE in Figure 7, suggesting that the equatorial ascent is indeed dominated by convection. By contrast, maximum potential ascent for nonconvective MAPE reaches local maxima in the extratropics.

The maximum potential ascent for nonconvective local MAPE is also compared to the climatological frequency of WCB starting points in Figure 7. WCB starting points are identified as the first point of the 48-hour ascent in the climatology of Madonna et al. (2014), and their frequency is plotted as the percentage of 6-hourly timesteps in which a WCB originates at a given location. The maximum potential ascent based on the climatological mean temperature and humidity is comparable in magnitude to, but somewhat smaller than, the WCB ascent threshold of 600 hPa, but larger values of maximum potential

ascent can occur on shorter timescales as shown below. The regions of higher maximum potential ascent in the nonconvective MAPE of the climatological atmosphere are very similar to the regions of heightened WCB activity, albeit with some regions of mismatch in the Western Pacific. Thus, the moist MAPE framework of Lorenz (1978, 1979) applied locally and excluding convection can identify regions of WCB activity based only on climatological mean temperature and humidity.

## 5.2    Daily maximum potential ascent and WCBs

The ability of the local nonconvective MAPE of the climatological atmosphere to identify regions of climatological WCB activity raises questions about the ability of this approach to predict WCB activity from temperature and humidity on shorter timescales. To investigate this, the local nonconvective MAPE and reference pressures are calculated using mean temperature and humidity values on individual days (the average of four six-hourly fields), and compared to the WCB activity over the same days. We refer to this as MAPE since it is based on the mean of temperature and humidity fields over a day, but it could

also be thought of as simply the daily available potential energy. Only one randomly-chosen year, 1985, is considered due to computational expense. Figure 8 shows results for example days near the beginning of the four seasons. Grid cells with one or more WCB starting points (it is not uncommon for multiple WCBs to start at the same location on a given day) tend to fall in regions of large maximum potential ascent. However, large maximum potential ascent does not always lead to WCB starting points on a given day because WCB activity also requires the presence of an extratropical cyclone on that day. There is

considerable spatial structure to the maximum potential ascent on a given day suggesting that extratropical cyclones themselves may be contributing to the spatial structure at this timescale.

To illustrate the statistical relationship between maximum potential ascent and WCB formation, the maximum potential ascent in daily local nonconvective MAPE calculations is calculated for each grid cell and each day in 1985, and these values are then linearly interpolated to the same grid as the WCB climatology. To properly compare to the climatology, which only

considers cyclones poleward of 20 degrees, we also only consider grid cells poleward of 20 degrees. The median of the maximum potential ascent over grid cells and days where at least one WCB starting point occurs is 570 hPa, which is substantially





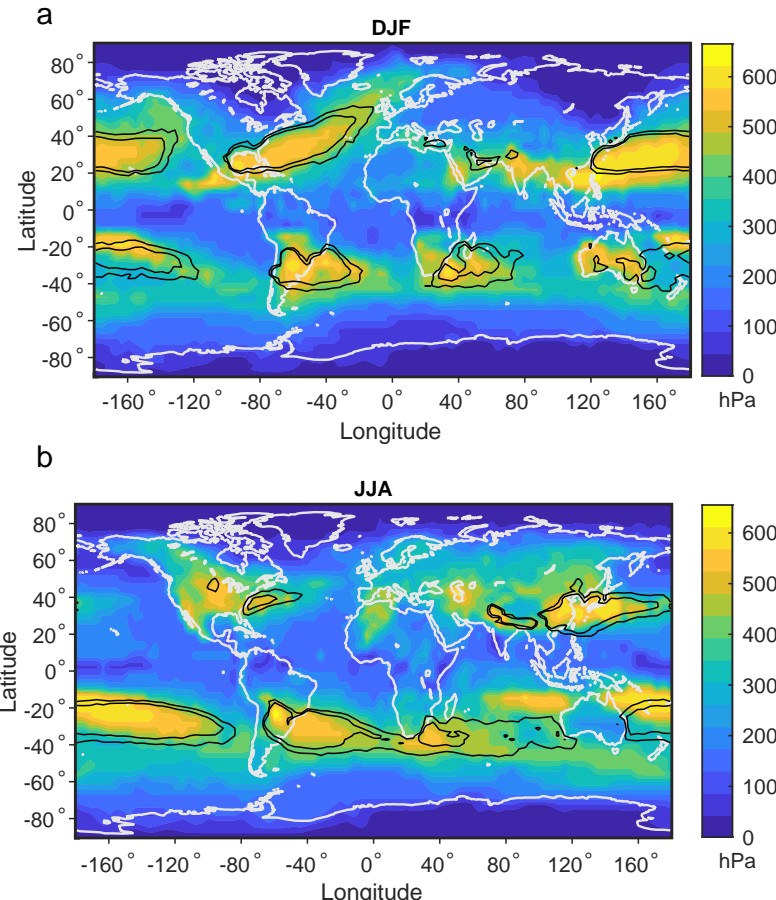

**Figure 7.** Maximum potential ascent in local nonconvective MAPE based on the climatological mean over 1979-2018 (color shading) and WCB starting point frequency over 1979-2018 (black contours of 1% and 2%) for (a) DJF and (b) JJA. Nonconvective MAPE is calculated as in Figure 5. Maximum potential ascent is calculated at each surface location as the maximum of pressure minus reference pressure for any parcel in the column at that surface location over all the nonconvective MAPE calculations that include that location. The WCB starting point frequency is shown as the percentage of six-hourly time steps in which a WCB begins at a given location in the climatology of Madonna et al. (2014)

higher than the overall median of 330hPa, and Figure 9a shows that the probability density functions of maximum potential ascent are quite distinct for these two cases. Figure 9b shows the probability of WCB formation for grid cells and days that exceed a given level of maximum potential ascent. 10% of grid cells with maximum potential ascent above 600 hPa have at 340 least one WCB starting point, and 25% of grid cells with maximum potential ascent above 720 hPa have at least one WCB starting point. These percentages may be compared with the baseline of 2% of grid cells over the extratropics (20° to 90°) that have at least one WCB starting point. Furthermore, the probability of WCB formation increases monotonically and very





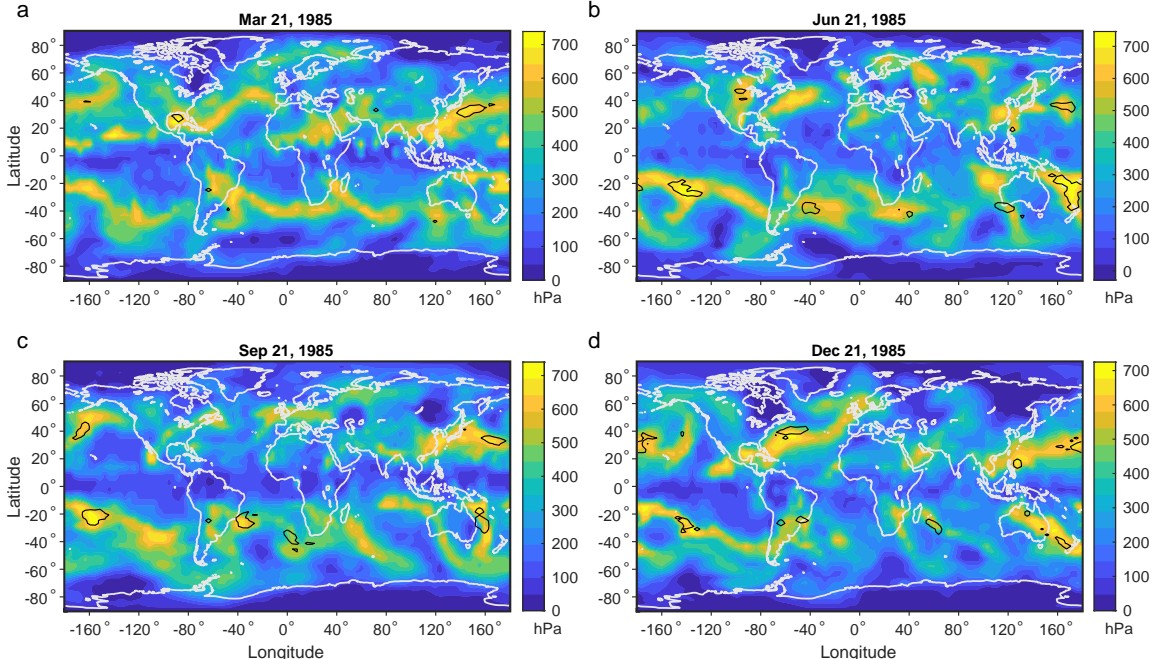

**Figure 8.** Maximum potential ascent in the local nonconvective MAPE (color shading) and WCB starting points (black contours) for example days in the year 1985. The black contours surround areas of at least one WCB starting point on the given day in the dataset of Madonna et al. (2014).

nonlinearly with maximum potential ascent. The probability of WCB formation based on high maximum potential ascent is even higher in storm track regions. For example, in a region of the Pacific from 160° to 190° in longitude and 40° to 60° N 345 in latitude over which extratropical cyclones occur with roughly 30% frequency in the climatology of Wernli and Schwierz (2006), roughly 50% of days and grid cells in 1985 with maximum potential ascent exceeding 700 hPa have at least one WCB starting point, compared to 8% probability of WCB starting points for all days and grid cells in that region in 1985.

The results above indicate that maximum potential ascent from daily local nonconvective MAPE calculations can predict with some skill the formation of WCBs using only the regional temperature and humidity fields. In conjunction with in-350 formation on cyclone occurrence, this maximum potential ascent may have value in understanding and forecasting of WCB formation, as well as understanding of the potential effects of climate change on WCBs.

## 6  Discussion and Conclusions

We have calculated the three-dimensional moist MAPE of the atmosphere for the first time, and we have also introduced a new approach to calculating a local MAPE on the characteristic length scale of extratropical cyclones. Combined with a





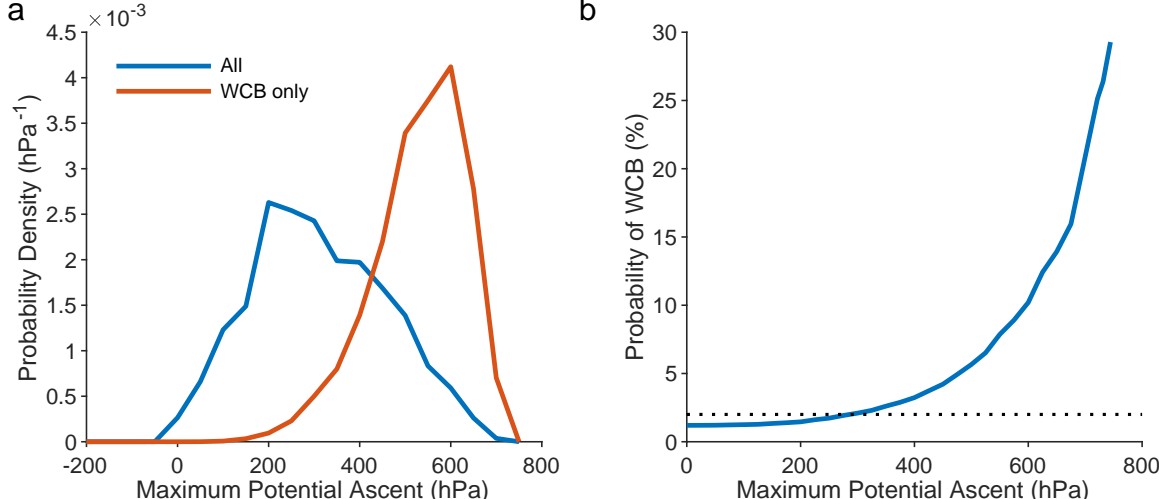

**Figure 9.** Statistical relation of WCB starting points to maximum potential ascent from daily local nonconvective MAPE calculations in the year 1985. (a) Probability density functions of the maximum potential ascent for all grid cells and days (blue) and for grid cells and days restricted to incidences of at least one WCB starting point (red). (b) Probability of at least one WCB starting point for grid cells and days above a given maximum potential ascent. For comparison, the probability of at least one WCB starting point for all days and grid cells in the extratropics (20° to 90°) is 2% (dotted line).

decomposition of MAPE into convective and nonconvective components, we show that MAPE provides a number of useful connections between the mean state of the atmosphere and the dynamics of extratropical cyclones.

    By including gradients of temperature and moisture in the zonal direction and by better accounting for topography which is difficult to deal with in a zonal-mean calculation, three-dimensional moist MAPE provides a more accurate accounting of the energy available to circulations as compared to previous calculations based on the zonal mean. Local three-dimensional

MAPE identifies regions where baroclinic eddies grow, broadly consistent with the Eady growth rate, but including the effects of latent heating and giving an energetic measure which could be useful for connecting to eddy kinetic energy. Differences between pressure and reference pressure in the local MAPE calculation characterize the thermodynamic potential for ascent. The maximum potential ascent is largest in the tropics for the full MAPE but in midlatitudes for the non-convective MAPE. The convective component of the local MAPE (i.e. the difference in the full and nonconvective MAPE) has some similarity in

terms of spatial pattern to a high percentile of CAPE based on instantaneous vertical profiles. Unlike high percentiles of CAPE, however, convective MAPE does not have a maximum near the equator, and this may be because it only measures available energy for convection driven by large-scale circulations and not convection driven by surface fluxes. Future work could investigate the extent to which there are connections between changes in EKE and local MAPE and changes in instantaneous CAPE and convective MAPE under climate-change scenarios.

We have also demonstrated that the maximum potential ascent in the calculation of nonconvective local MAPE is related to WCB formation. This is a new connection between the mean state of the atmosphere and important high-frequency weather





events. In particular, WCBs are crucial components of extratropical cyclones and every-day weather in the midlatitudes, and they plan an important role in cloud formation, precipitation, and transport of air pollution (Browning, 1990; Stohl et al., 2002; Pfahl et al., 2014). The maximum potential ascent based on the climatological mean temperature and humidity skillfully
identifies regions of heightened WCB formation, and this link could be used to better understand the geographical and seasonal distribution of WCBs. Furthermore, regions of high maximum potential ascent on individual days are more likely to form WCBs than other regions. It is possible that the link between maximum potential ascent and WCB formation on the daily timescale could be helpful for forecasting of WCBs, but establishing this would require a detailed study of statistics such as the conditional probability of the occurrence of a WCB given the presence of both a cyclone and strong potential ascent versus
only the presence of a cyclone. In addition there may be a useful link between maximum potential ascent and WCB formation on longer subseasonal timescales. Two other areas of future work related to maximum potential ascent seem promising. First, it could be investigated whether there is a simple analytical approximation for the maximum potential ascent as a function of mean temperature and humidity fields. Second, it would be interesting to investigate changes in maximum potential ascent in simulations of climate change to provide a link between mean warming and moistening of the atmosphere and changes in
WCB formation.

*Code and data availability.*   A directory including all analysis code used in this study is published online at *https://doi.org/10.5281/zenodo.5826260*

## Appendix A: Uniform grid generation

We create a uniform grid on the sphere following the methodology of Rosca (2010). The strategy used is to create an equally spaced grid on a square, and then to project it onto each hemisphere by using area-preserving bijections that map from a square
to a disk and then from the disk to a hemisphere. The square is of edge $e$, and the sphere of radius $r$, where $e = r\sqrt{2\pi}$. Further details and proof of this methodology can be found in Rosca (2010).

The grid on the square is created using an identical set of Cartesian coordinates in x and y. For convenience, $L$ is defined such that $e = 2L$. The number of grid cells in each direction is denoted $N$. The spacing, $d$, is then $e/N$. The grid cells are centered at grid points starting at $-L+d/2$ and extending with a spacing of $d$ to $L-d/2$. To project this grid onto the Southern Hemisphere
and preserve area, we then apply the following formulas to project $(a,b)$, a set of $x$ and $y$ coordinates, into $(A,B,C)$, a set of $x$, $y$, and $z$ coordinates.

1. For $0 \leq |b| \leq |a| \leq L$,

$$(A, B, C) = \left( \frac{2a}{\pi} \sqrt{\pi - \frac{a^2}{r^2}} \cos \frac{b\pi}{4a}, \ \frac{2a}{\pi} \sqrt{\pi - \frac{a^2}{r^2}} \sin \frac{b\pi}{4a}, \ \frac{2a^2}{\pi r} - r \right); \tag{A1}$$

2. For $0 \leq |a| \leq |b| \leq L$,

$$(A, B, C) = \left( \frac{2b}{\pi} \sqrt{\pi - \frac{b^2}{r^2}} \sin \frac{a\pi}{4b}, \ \frac{2b}{\pi} \sqrt{\pi - \frac{b^2}{r^2}} \cos \frac{a\pi}{4b}, \ \frac{2b^2}{\pi r} - r \right). \tag{A2}$$



For the Northern Hemisphere, the formulas are the same as in Equations A1 and A2, except that *C* is of opposite sign. These Cartesian coordinates can be easily transformed to latitude and longitude on the sphere of radius *r*.

Figure 1 shows an example of such a grid which we employ in the analysis and which has $40 \times 40$ grid cells in each hemisphere.

## Appendix B: Sensitivities of calculation of local MAPE

We focus on the calculation of local MAPE as an example, and we describe its sensitivity to resolution, assumed eddy size and calculation method.

### B1  Sensitivity to resolution

Low horizontal resolutions are not sufficent to capture the characteristic patterns of local MAPE (Figure B1a) or maximum potential ascent (Figure B2a). However, at resolutions at and beyond roughly 1800 surface grid cells (a $30 \times 30$ grid in each hemisphere) and at and beyond ten vertical levels, the results converge (Figures B1b,c and B2b,c).

### B2  Sensitivity to the eddy size

Unlike for horizontal resolution, as the radius of the subdomains used to calculate local MAPE increases, the results do not converge. Indeed, one would expect the magnitude of MAPE to increase as the subdomain increases in size, consistent with the scaling of dry zonal-mean MAPE per unit mass with the square of the length of the domain considered (O'Gorman and Schneider, 2008). However, the spatial pattern of local MAPE does appear to converge for geodesic radii greater than roughly 10 degrees even though the MAPE value continues to increase as the radius increase (Figure B3). Interestingly, the spatial pattern of maximum potential ascent seems to converge at a larger radius, with noticeable differences between the 10 degree and 15 degree calculations, but similar spatial patterns of maximum potential ascent for the 15 degree and 20 degree calculations (Figure B4).

### B3  Sensitivity to the method of calculation

In most of the atmosphere, and in particular in nearly all of the extratropics, the divide-and-conquer algorithm and integer linear programming give nearly identical values of local MAPE and maximum potential ascent (Figure B5 and Figure B6). However, in some regions in the tropics (where the MAPE is small), the divide-and-conquer algorithm can be very inaccurate as a fraction of the exact MAPE given by integer linear programming. Indeed, the divide-and-conquer algorithm sometimes even returns a negative value for MAPE (black contours in Figure B5), which is physically impossible but has been previously shown to sometimes occur for certain conditionally unstable columns with the divide-and-conquer algorithm (Harris and Tailleux, 2018). Figure B5 demonstrates this issue in the tropics for DJF and differences are similar across all seasons. Similarly, in some areas in the tropics, the exact calculation predicts large maximum potential ascent that is not captured by the divide-and-conquer



algorithm (Figure B6). These inaccuracies of the divide-and-conquer algorithm do not affect the main conclusions of this

paper.





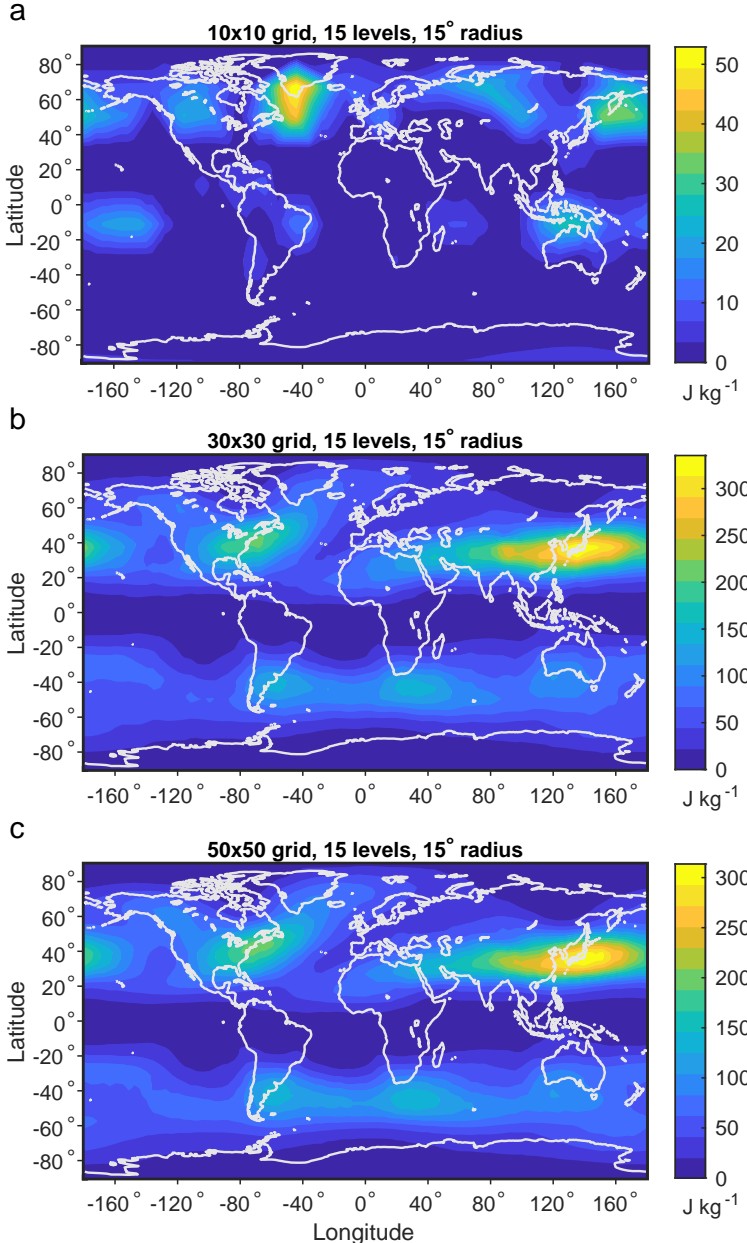

**Figure B1.** Local MAPE based on the DJF climatological mean over 1979-2018 for grids of varying horizontal resolution: (a) 200 surface grid cells (a $10 \times 10$ grid in each hemisphere), (b) 1800 surface grid cells (a $30 \times 30$ grid in each hemisphere), and (c) 5000 surface grid cells (a $50 \times 50$ grid in each hemisphere). All results shown in this figure use 15 pressure levels and subdomains of geodesic radius 15 degrees (1670 kilometers).



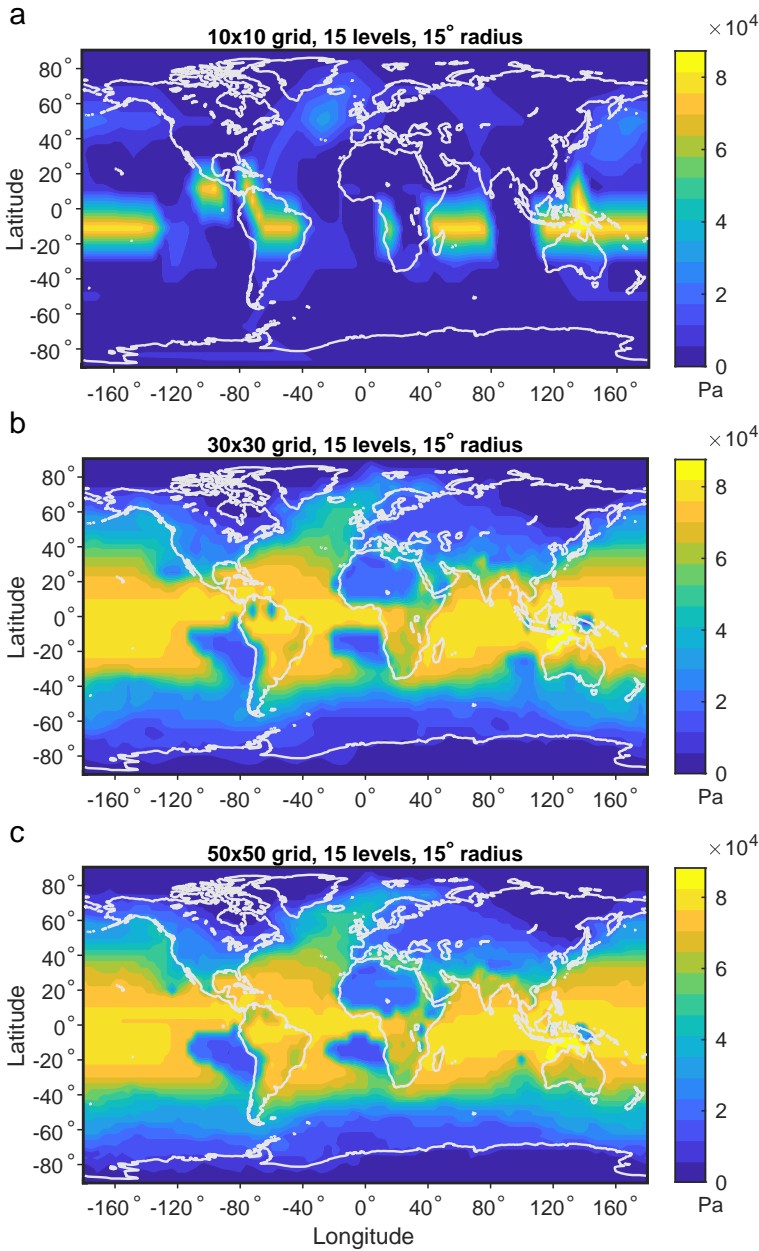

**Figure B2.** Maximum potential ascent in local MAPE based on the DJF climatological mean over 1979-2018 for grids of varying horizontal resolution: (a) 200 surface grid cells (a $10 \times 10$ grid in each hemisphere), (b) 1800 surface grid cells (a $30 \times 30$ grid in each hemisphere), and (c) 5000 surface grid cells (a $50 \times 50$ grid in each hemisphere). All results shown in this figure use 15 pressure levels and subdomains of geodesic radius 15 degrees (1670 kilometers).



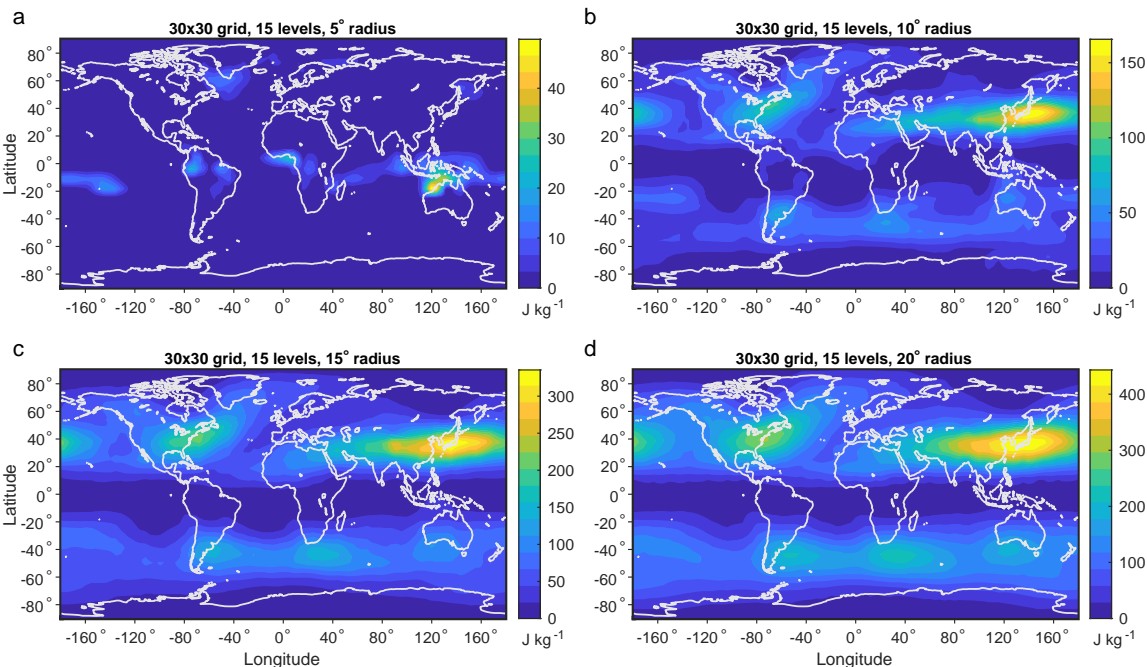

**Figure B3.** Local MAPE based on the DJF climatological mean over 1979-2018 using subdomains of varying geodesic radii: (a) 5 degrees (560 km), (b) 10 degrees (1110 km), (c) 15 degrees (1670 km), and (d) 20 degrees (2230 km). All results in this figure use 1800 surface grid cells (a $30 \times 30$ grid in each hemisphere) and 15 pressure levels.



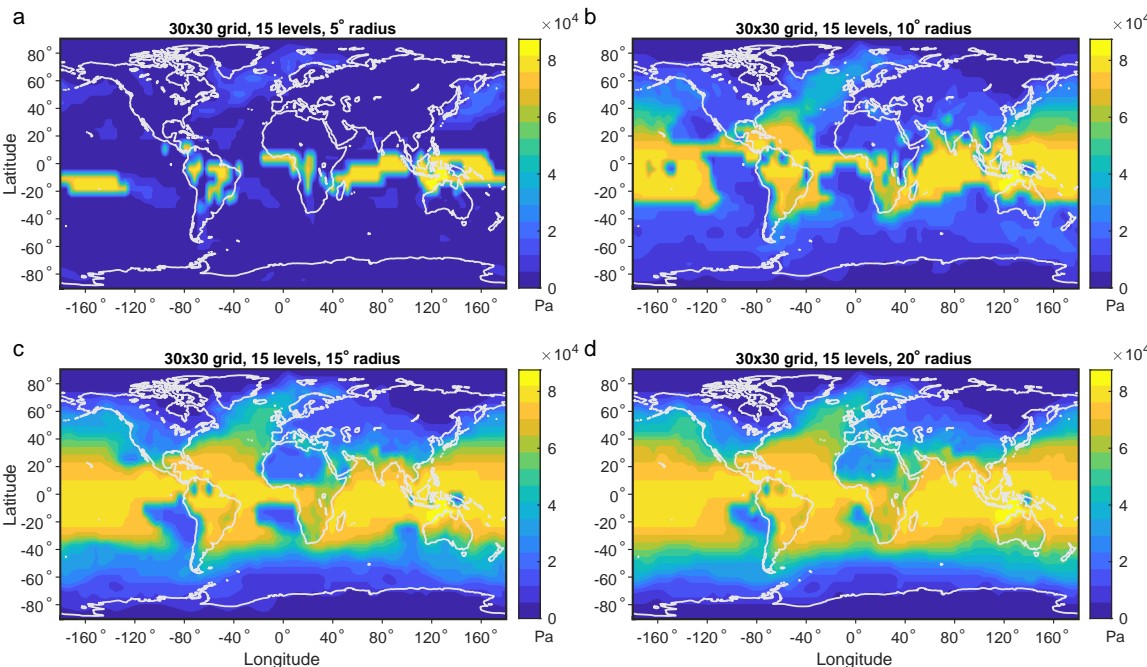

**Figure B4.** Maximum potential ascent in local MAPE based on the DJF climatological mean over 1979-2018 using subdomains of varying geodesic radii: (a) 5 degrees (560 km), (b) 10 degrees (1110 km), (c) 15 degrees (1670 km), and (d) 20 degrees (2230 km). All results in this figure use 1800 surface grid cells (a $30 \times 30$ grid in each hemisphere) and 15 pressure levels.





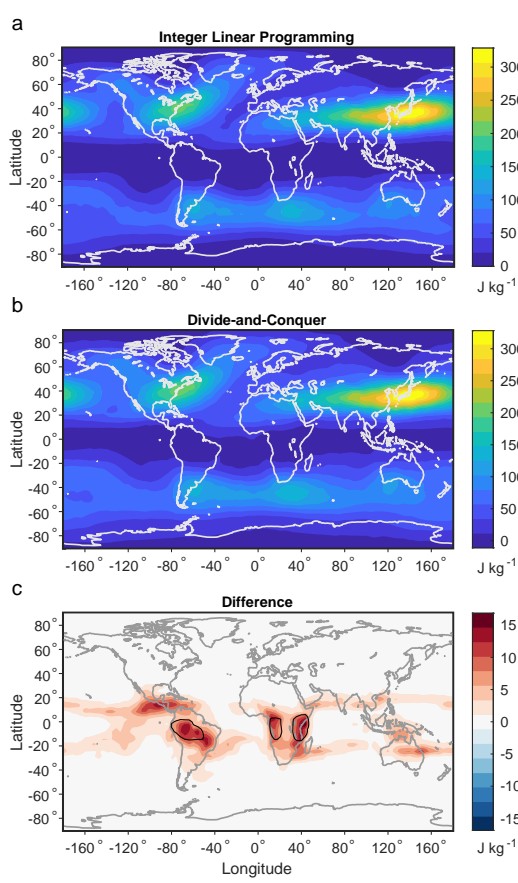

**Figure B5.** Sensitivity to calculation method for the local MAPE based on the DJF climatological mean over 1979-2018. Shown is MAPE calculated using (a) the exact integer linear programming approach, (b) the approximate divide-and-conquer alogrithm, and (c) the difference reported as integer linear programming minus divide and conquer. Black contours in (c) indicate regions where the divide-and-conquer algorithm gives negative values.



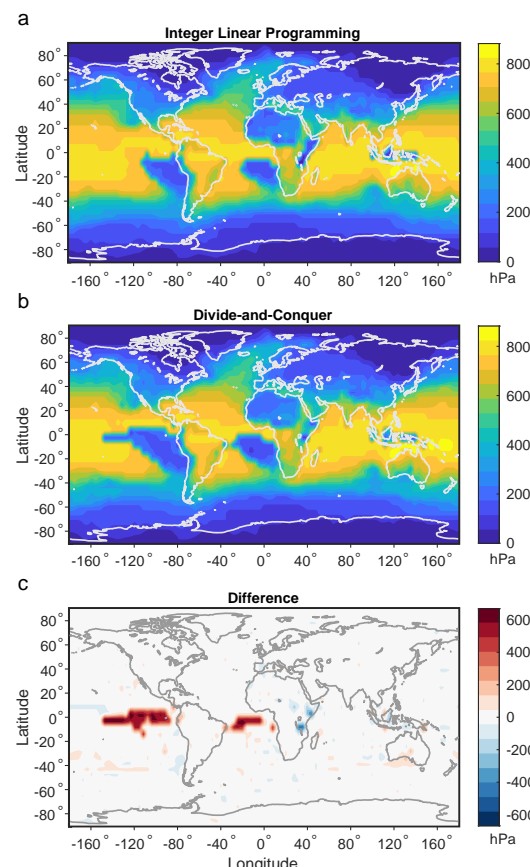

**Figure B6.** Sensitivity to calculation method for the maximum potential ascent in local MAPE based on the DJF climatological mean over 1979-2018. Shown is maximum potential ascent calculated using (a) the exact integer linear programming approach, (b) the approximate divide-and-conquer alogrithm, and (c) the difference reported as integer linear programming minus divide and conquer.





*Author contributions.* C.G.G., P.A.O'G., and S.P. conceived of and designed the research and contributed to the editing of the text. C.G.G. performed the research, produced all figures, and drafted the paper.

*Competing interests.* S.P. serves as executive editor of Weather and Climate Dynamics.

*Acknowledgements.* We are grateful to H. Wernli, for helpful discussions and for first pointing out the possibility of a connection between WCBs and parcel rearrangements in MAPE. We are also grateful to R.G. Prinn for helpful insight. C.G.G. was supported by the Industry and Foundation sponsors of the MIT Joint Program on the Science and Poilicy of Global Change, NASA grant NNX16AC98G to MIT, and the National Science Foundation Graduate Research Fellowship Program under NSF Grant 1122374. P.A.O'G. acknowledges support from NSF Grant AGS 1749986. S.P. acknowledges support from Deutsche Forschungsgemeinschaft (DFG) through grant CRC 1114 "Scaling Cascades
in Complex Systems", Project Number 235221301, Project (C06).



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
