# Peer review of "Moist available potential energy of the mean state of the atmosphere and the thermodynamic potential for warm conveyor belts and convection"

_EGUsphere, 2022_

## Author Response (AR1)

**Author's response Moist available potential energy of the mean state of the atmosphere and the thermodynamic potential for warm conveyor belts and convection**

Below please see our responses to both referee comments, in order, with referee comments reproduced for ease of comprehension.

**Referee # 1:**

**Summary:**

The authors calculate 2d maps of mean/moist available potential energy (MAPE) from timeaveraged reanalysis data. They then partition the MAPE in two different ways: 1) Non-convective MAPE is found by restricting the vertical rearrangement of parcels, and the remaining MAPE is convective MAPE, and 2) A local MAPE is calculated by restricting the distance over which the parcels can be rearranged. The authors then connect the non-convective local MAPE, and associated ascent, with warm conveyor belts (WCB), including example instantaneous snapshots as well as time-averaged data.

The calculations of MAPE are well described and the partitioning into different components is interesting. However, the interpretation of the results, in particular the explanations in terms of CAPE and WCBs are lacking in depth and insight. This could be a very good paper, and the calculations of MAPE are interesting in their own right, but to justify some of the statements in the abstract and having "warm conveyor belts" in the title, further work is needed. So, I have recommended major revisions

**Major Comments:**

The connection of convective MAPE to a high percentile of instantaneous CAPE looks weak. Is there a reason you didn't also calculate CAPE from the mean state? In DJF the contour chosen seems to somewhat match the regions of high MAPE, but since only one value is contoured it looks like the value has been chosen to fit rather than an actual correspondence. It would be useful to show more values in the contours.

In JJA the MAPE has very little correspondence to the contour shown and looks more like it picks out regions of more tropical/convective storms: western pacific and Atlantic tropical storms and the Indian monsoon. I think the DJF map could also have similar explanations. My knowledge of Southern Hemisphere meteorology is not so good, but the local maxima seem like they could also relate to tropical storm regions. Also, there appears to be a strong signature of the African Easterly Jet in DJF. It may be useful to explain the convective MAPE in this way rather than just in terms of CAPE. Rather than connecting MAPE to "instantaneous atmospheric convection", it could be connected with convectively driven storms.

The statement in the abstract that the maximum potential ascent in the MAPE calculation "skilfully identifies the necessary conditions for WCBS" is misleading. All that has really been shown is that there is some correlation between WCB genesis regions and the ascent in the MAPE calculation. To me, it looks like the MAPE picks out the storm tracks, and because WCBs are associated with storms there is some relation.

The idea that WCBs will relate to ascent in the MAPE calculation makes sense as they are the ascending air in extratropical cyclones and therefore will relate to this instability but the results don't show that MAPE is adding any value to this. The authors state that they interpret the non-convective local MAPE as energy available for the "generation of large-scale eddies through moist baroclinic instability" and show that it has a similar pattern to the Eady growth rate. So my question is what does value does MAPE add over the Eady growth rate (which is much easier to calculate) in predicting/explaining warm conveyor belts? I wonder if this could be shown by relating the ascent predicted in the MAPE calculation with the actual ascent in the WCB trajectories.

I would also like some discussion of the large areas where there is ascent predicted by the MAPE calculation, and presumably large Eady growth rate, but no warm conveyor belts. Presumably this is just related to where cyclones do and don't actually form, but if the MAPE calculation or some additional variable can't predict this then I don't see how it can be described as skilfully predicting warm conveyor belts.

**Minor Comments:**

•P3L60 – "More than half of extratropical cyclones are associated with a WCB in northern hemisphere winter...". This seems very low. I would expect most cyclones to be associated with a WCB. One reason this could be so low is that the Madonna et al. (2014) climatology uses the strict 600 hPa criteria for identifying WCB trajectories and so miss weaker WCBs. Madonna et al. (2014) acknowledge that they are not aiming to identify the full WCB airmass in their climatology and test the sensitivity of using a 500 hPa threshold instead. As far as I can tell, Madonna et al. (2014) doesn't go into quantifying the co-occurrence of WCBs with cyclones, so I don't know where this number, or the other details about WCBs and cyclones in the sentence, has come from.

• Figure 2 – The caption states that (a,b) shows 796 hPa but the figure says 864 hPa.

**Author response to referee #1:**

Thank you for these very thoughtful comments, which we feel will improve the manuscript greatly.

The first major comment regarding the connection of convective MAPE to high percentile CAPE inspired an additional direction which we feel strengthens the paper. We have removed the discussion of high percentile CAPE, and we instead added discussion and analysis that compares convective MAPE with the occurrence of intense convective events using a precipitation feature database from the Global Precipitation Mission, where we found notable correspondence. Please see the new title of the paper and updated Figure 6 and discussion of that figure in section 4.3.2 of the new manuscript.

In response to the second major comment regarding the skillful identification of the necessary conditions for WCBs, we have adjusted the statement in the abstract to read "This maximum potential ascent can be calculated based only on mean temperature and humidity, and WCB's tend to start in regions of high maximum potential ascent on a given day." This is supported by Figure 8 which demonstrates that on daily timescales WCB starting points occur in the areas of maximum potential ascent. The maxium potential ascent is not simply picking out storm track regions, as there is spatial heterogeneity on daily timescales within the storm track. Furthermore the spatial and distributions and seasonal cycles of climatological nonconvective MAPE and Eady growth (Fig 5) are different from those of maximum potential ascent and WCB starting points (Fig 7). For example, these distributions are quite different in the Southern Hemisphere in DJF or in the Northern Hemisphere in JJA as now pointed out in section 5.1.

In response to the third major comment regarding what MAPE adds over the Eady growth rate: while the Eady growth rate is easier to calculate, there is value in the unified theory that MAPE provides in this context, which includes energetics and ascent. As a thermodynamic measure of magnitude of eddies, non-convective MAPE has advantages over Eady growth rate in that it (1) includes the effects of latent heating whereas the Eady growth rate is a dry quantity applied to a moist atmosphere, and (2) it provides an energetic measure which can be directly compared to eddy kinetic energy. These two points are made in section 4.3.1. Furthermore, we emphasize that the Eady growth rate does not adequately capture the seasonality or spatial distributions of WCB

starting points (e.g., compare the Southern Hemisphere in DJF in figure 5 versus figure 7). This is now emphasized at the end of section 5.1.

The reviewer asks if the ascent predicted by MAPE can be related to ascent in WCB trajectories. That is indeed the case: there is numerical consistency in the threshold of ascent used to define WCBs in the Madonna et al climatology (600hPa) and the values of maximum possible ascent for days with WCBs (see fig 9a; red curve) which have a median value of 570hPa as now emphasized in the text in section 5.2.

The reviewer asks for discussion of the large areas where there is ascent predicted by the MAPE calculation, but no WCBs. This is a helpful clarification – we consider the concept of maximum potential ascent similar to the idea of maximum potential intensity in tropical cyclones in the sense that the calculation reveals the thermodynamic potential for an event which may not necessarily be realized at all times. We have added language to this effect in section 5.2.

We have address both minor comments in the new manuscript as well. We have added a citation to better support the statement, "More than half of extratropical cyclones are associated with a WCB in northern hemisphere winter...", and we have corrected the caption on Figure 2.

**Referee #2:**

In this manuscript the authors extend the traditional view of MAPE into three dimensions, and examine its local variations. They further decompose MAPE to its non-convective and convective components and show that the former accounts for most of the total MAPE, and can be expressed via linear baroclinic theory, in the form of the Eady growth rate. Lastly, the authors show that the maximum potential ascent associated with non-convective MAPE is linked to WCB. Deriving local MAPE is an interesting exercise, and in the context of WCB, it seems that one could retrieve new physical understanding that links Eulerian and Lagrangian perspectives of the mid-latitude flow. It is unfortunate, in my opinion, that the authors do not further investigate such avenue to yield new physical understating of the system. Instead, through most of the paper, the authors focus on results which do not necessarily allow us to learn new physics on the mid-latitude flow, or simply describe how MAPE behaves spatially. Related to the above point, the paper is rather technical, and in several cases repetitive, and the writing is not concise; in several places this only diverts the reader from the main take-home message.

Major comments:

The authors show that mid-latitude MAPE, which follows non-convective MAPE, basically describes the baroclinic zones in the mid-latitudes, which one could also retrieve from Eady growth rate. Why is it thus necessary to thoroughly discuss the derivation of MAPE (separating its convective and non-convective components) and its spatial patterns? What new information have we acquired here on the mid-latitude flow? On the other hand, the results that links MAPE to WCB are interesting, as it allows us to learn how such events are created and how they are linked to the mean state in the atmosphere. My suggestion is to further explore this link, and provide a new piece of physical understating.

The introduction and method sections are considerably long, and include large amount of details. In my opinion this only diverts the reader from the main take home message as the reading becomes cumbersome. For example, in the introduction the authors not only discuss the results but also the methods. Furthermore, that exact algorithm used to calculate the mean state (e.g., divide-and-conquer), is an unnecessary detail in my opinion. There are other examples of that throughout the method section.

Although the authors chose to show results from both DJF and JJA, the discussion on Figs. 3-7 is almost entirely limited to DJF. Either remove panels which you do not discuss, or extend your discussion to JJA as well. Specifically, why does the structure of MAPE in JJA does not follow that of DJF? Why does MAPE maximizes in land?

**Minor comments:**

- You chose to analyze Era-Interim, what about other reanalysis products? How do you know that your results are not product dependent?

- In several locations throughout the manuscript the term "northern/southern hemisphere" is lacking capital letters (e.g., lines 58-59).

-line 148: recognize - > recognized

**Author response to referee #2:**

Thank you for these insightful comments, which have led to changes that we feel improve the manuscript.

Like Reviewer 1, Reviewer 2 asked about advantages of MAPE over the Eady growth rate. We highlight that the MAPE calculations presented in this paper offer a unified framework that connects to multiple weather aspects (please see reply on RC1 for further discussion). We add that the Eady growth rate is not an energetic measure, and it does not take latent heating into account, both of which are noted in section 4.3.1 of the manuscript. Also, the Eady growth rate does not provide information about occurrence of WCB starting points which have a distinct spatial pattern and seasonality compared to the Eady growth rate (compare figures 5 and 7, e.g. in the Southern Hemisphere in DJF) as now emphasized in the text in section 5.1. Lastly, the Eady growth rate does not give us information about convection, whereas we find that convective MAPE has some correspondence to regions of intense convection (see the revised figure 6).

We agree with the reviewer that it would be worthwhile to further develop the link from MAPE maximum potential ascent to WCBs, especially by developing a simple analytical expression for maximum potential ascent (akin to the Eady growth rate but predicting maximum potential ascent instead of growth). We have mentioned this possibility in the conclusions section, but it is non-trivial to develop such a theoretical expression and beyond the scope of the present work.

The reviewer suggested helpful ways to simplify the paper, and we have made changes that we feel improve the readability of the paper significantly. We have moved the discussion of algorithmic approaches to MAPE calculations from the introduction to the methods section 3 and shortened it. We have also added a sentence to direct readers who are only interested in the results before the description of the exact approach to calculating MAPE (just prior to section 3.1).

The reviewer also asks about why we chose to show both DJF and JJA in Figs. 3-7. We think it is important to show both DJF and JJA seasons because it helps to show correspondence between different aspects of MAPE and features such as WCBs. Furthermore, it would be confusing to only show the results for one season for some aspects and both seasons for others. We added text to section 4.2 to point out that the locations of high local MAPE are different in summer versus winter due to the evolution of land-ocean contrasts in horizontal temperature gradients and static stability over the seasonal cycle.

The reviewer also asked about our choice of ERA-Interim as a reanalysis product. We chose ERA-Interim reanalysis as it was regarded as the best product at the time the analysis was done. The fact that we are relying on only monthly data (or daily data in one section) rather than subdaily data, and the agreement in major features in midlatitude energetics described in O'Gorman (2010), which uses NCEP2 reanalysis, and Gertler and O'Gorman (2019), which uses ERA-Interim reanalysis allows us to feel confident that our results are not product dependent. We have addressed the typos identified by the reviewer in the new manuscript.

---

## Author Response (AR2)

Author Response
**Moist available potential energy of the mean state of the atmosphere and the thermodynamic potential for warm conveyor belts and convection**

Minor corrections made to final manuscript:

1. Typos fixed on lines 248, 314, and 412
2. Line 10: "nonconvective local MAPE" changed to "local nonconvective MAPE" for consistency with rest of abstract.
3. Figure 6 Caption: "Intense convection events are calculated using GPM" changed to "Intense convection events are identified using GPM" for accuracy.